

# Simulating natural carbon sequestration in the Southern Ocean: on uncertainties associated with eddy parameterizations and iron deposition

Heiner Dietze[1], Julia Getzlaff[1], and Ulrike Löptien[1]

[1]GEOMAR - Helmholtz Centre for Ocean Research Kiel

*Correspondence to:* Heiner Dietze (hdietze@geomar.de), Ulrike Löptien (uloeptien@geomar.de)

**Abstract.** The Southern Ocean is a major sink for anthropogenic carbon. Yet, there is no quantitative consensus about how this sink will change when surface winds increase (as they are anticipated to do). Among the tools employed to quantify carbon uptake are global coupled ocean-circulation biogeochemical models. Because of computational limitations these models still fail to resolve potentially-important spatial scales. Instead, processes on these scales are parameterized. There is concern that

deficiencies in these so-called *eddy-parameterizations* might imprint wrong sensitivities of projected oceanic carbon uptake. Here, we compare natural carbon uptake in the Southern Ocean simulated with contemporary eddy-parameterizations. We find that very differing parameterizations yield surprisingly similar oceanic carbon in response to strengthening winds. In contrast, we find (in an additional simulation) that the carbon uptake does differ substantially when the supply of bioavailable iron is altered within its envelope of uncertainty. We conclude that a more comprehensive understanding of bioavailable iron dynamics

will substantially reduce the uncertainty of model-based projections of oceanic carbon uptake.

## 1 Introduction

More than two decades after the discovery of major glacial/interglacial cycles in the $CO_2$-concentration of the atmosphere, it is believed that no single mechanism can account for the full amplitude of past $CO_2$-variability (e.g. Sigman and Boyle, 2000). There is, however, growing evidence that the variability in the extent, to which deep-water masses are isolated from the

atmosphere in the Southern Ocean, is among the major drivers regulating atmospheric $CO_2$-variability. In this context, the role of wind-driven upwelling is of special interest (e.g. Lenton and Matear, 2007; Lovenduski et al., 2007; Marshall and Speer, 2012), especially since Anderson et al. (2009) linked increased ventilation of deep water to the deglacial rise in atmospheric $CO_2$.

There is also evidence that wind-driven upwelling will shape the future evolution of atmospheric $CO_2$-concentrations: ob-

servations during the recent decades show a strong upward trend of the dominant mode of climate variability in the Southern Hemisphere (Marshall, 2003), which is, most likely, driven by increased greenhouse gas concentrations. This upward trend of the so-called *Southern Annular Mode* is related to stronger surface winds and a pole-ward shift of the westerlies - and - is projected by climate scenarios to intensify (e.g. Simpkins and Karpechko, 2012). As to how the projected wind changes will quantitatively link to upwelling of deep waters with high carbon content (which in turn affects atmospheric $CO_2$-concentrations) is,



however, not comprehensively understood. The current generation of coupled ocean-circulation biogeochemical models still struggles to retrace observed trends (Lenton et al., 2013) and the models differ considerably as regards their representation of anthropogenic carbon in the Southern Ocean (Frölicher et al., 2015).

For now we know that the Southern Ocean (here defined as the region south of $40°$S) accounts for more than 40% of the total annual oceanic $CO_2$-uptake (Takahashi et al., 2009). Further, there is evidence, based on inversions of atmospheric $CO_2$-concentrations (Le Quérér et al., 2007) and trends in the difference between partial pressures of $CO_2$ in the surface ocean and the atmosphere (Metzl, 2009; Takahashi et al., 2009) that the uptake of $CO_2$ in the Southern Ocean declines.

The link between strengthening winds and a declining Southern Ocean carbon sink is, however, inconclusive: global ocean-carbon models driven by observed wind patterns suggest that increased winds drive an increased exposure of carbon-rich deepwater to the surface. This leads to an overall reduced gradient between the atmosphere and the surface ocean and, subsequently, to a decreased oceanic $CO_2$-uptake. More specifically, all state-of-the-art coarse resolution ocean models suggest that the enhanced equatorward Ekman transport associated with a poleward shift and intensification of the southern hemisphere westerlies results in an increased circulation in the subpolar meridional overturning cell (e.g. Saenko et al., 2005; Hall and Visbeck, 2002; Getzlaff et al., 2016)). This implies an increased upwelling of deep water, rich in dissolved inorganic carbon, south of the circumpolar flow (e.g. Zickfeldet al., 2007; Lenton and Matear, 2007; Lovenduski et al., 2008; Verdy al., 2007).

On the other hand, high resolution models that explicitly resolve eddies rather than parameterizing their effect, show that stronger westerlies induce an increased eddy activity and suggest that the associated eddy fluxes could compensate initial increases in northward Ekman transport (Hallberg and Gnanadesikan, 2006; Hogg et al., 2008; Screen et al., 2009; Thompson and Solomon, 2002). As a consequence, the upwelling, associated air-sea carbon fluxes, isopycnal tilt and the transport of the Antarctic Circumpolar Current should be rather insensitive towards changes in the wind forcing. In line with the high resolution models, observations by Argo floats do not reveal any changes in isopycnal tilt as a response to increasing winds (Böning et al., 2008). This suggests that the wind-induced upwelling can indeed be compensated by eddy fluxes.

To date, earth system models that are used to project air-sea carbon fluxes do not explicitly resolve eddy fluxes. Because of computational constraints the relevant spacial scales can not be resolved and the respective processes have to be parameterized. Typically, non-eddy resolving ocean models employ the parameterization of Gent and McWilliams (1990) (hereafter GM) to account for the effects of (unresolved) turbulent lateral advection. The "strength" of these effects in the GM parameterization is determined by a parameter, the so-called *thickness diffusivity*. In the past, for pragmatic reasons, the thickness diffusivity has been set to a globally constant value. On physical grounds, however, there is no justification for a global uniform thickness diffusivity. To the contrary, e.g. satellite observations and eddy resolving modelling clearly reveal that eddy activity varies strongly in space and time. This implies that the effect of eddies on the mean flow is inhomogeneously distributed over the ocean. Thus, recent advances have been aimed at taking the variability of the eddy field into account by parameterising the thickness diffusivity as a local function of e.g. stratification, *Eady growth rate*, or a combination of the Eady growth rate, the *Rossby radius* and the *Rhines scale*. The results of these advances towards a more realistic closure for the thickness diffusivity clearly show that mean (resolved) properties such as e.g. the simulated strength of the Antarctic Circumpolar Current or the Meridional Overturning circulation are sensitive towards the choice of the closure (Eden et al., 2009; Viebahn, 2010). This



suggests that the simulated upwelling of deep carbon-rich waters in the Southern Ocean may as well be sensitive to the choice of the closure.

In the present study, we test differing closures for GM's thickness diffusivity in a global coarse-resolution coupled ocean-circulation biogeochemical model (which comprises carbon). The focus is on how the closures will affect the sensitivity of

carbon uptake in response to increasing winds in the Southern Ocean. To this end, we will revisit the apparently discrepant responses to trends in the Southern Annular Mode of (1) coarse resolution models on the one hand, and (2) eddy-resolving models and observations on the other hand.

In order to put our results concerning uncertainties associated with physical processes into perspective, we compare it with the uncertainty that is associated with uncomprehensively-understood deposition of bioavailable iron.

The following Section 2 describes our global coupled ocean-circulation biogeochemical model configurations and the respective simulations. The subsections 2.2.1 and 2.2.1 give a short introduction to eddy-parameterizations and iron, respectively. In Section 3 we compare all simulations with one another. The focus is on the simulated carbon uptake of the Southen Ocean and related processes. The paper ends with a conclusive summary in Section 4.

## 2 Model

This study is based on simulations with the Modular Ocean Model (MOM), version MOM4p1. Specifically we use the ocean-ice component of the CM2Mc configuration coupled to the **B**iology **L**ight **I**ron **N**utrients and **G**asses (BLING) ecosystem model of (Galbraith et al., 2010). We force with climatological atmospheric conditions (*Normal Year Forcing* of Large and Yeager (2004)). Our configuration is identical to the one used and described in Galbraith et al. (2010).

The nominal zonal resolution is $3°$. The meridional resolution varies from $3°$ in mid-latitudes to $2/3°$ near the equator.

Additional regions of enhanced meridional resolution are the latitudes of the Drake Passage and respective latitudes on the northern hemisphere. In the Arctic, a tri-polar grid is applied to avoid discontinuities at the North Pole (c.f. Griffies et al., 2005). The vertical discretisation comprises 28 levels with a resolution ranging from 10 m at the surface to 506 m at depth.

In total we perform three 1000 year-long model spin-ups: one that is identical to the configuration described in Galbraith et al. (2010), and yet another two that differ - as we will elaborate on in Section 2.1 - as regards the parameterization of unresolved

eddies (i.e. thickness diffusivity, - after Gent and McWilliams, 1990). All three spin-ups start from the semi-equilibrated state of Galbraith et al. (2010). In all simulations the atmospheric $CO_2$-concentration is prescribed to a pre-industrial level of 278 ppmv. This choice of oceanic carbon boundary conditions is often applied to study the oceanic uptake of *natural* (as opposed to anthropogenic) carbon uptake (e.g. Lovenduski et al., 2007). The results from the spin-ups are evaluated in appendix A.

Each of the three spin-ups is extended in order to assess the sensitivity of simulated natural carbon uptake in the Southern Ocean towards anticipated wind changes. In each of the extensions the magnitude of the wind speeds south of $40°$S is increased by a rate of 14% in 50 years. This increase is consistent with results from reanalysis for the period 1958 to 2007 (Lovenduski et al., 2013).





In the remainder of this Section we describe our sensitivity experiments in more detail. These experiments explore the impact of differing eddy parametrizations and compare it to uncertainties that are related to biogeochemical responses to changes in air-sea deposition of bioavailable iron. Tab. 1 summarizes the experimental setup. The following sub-section are organized as follows:

– Section 2.1 starts with an overview of approaches to parameterize the effect of eddies in coarse resolution models (Section 2.1.1), followed by an explicit description of our model setups and the respective simulations with differing eddy-parameterizations in Section 2.1.2.

– Section 2.2 starts with an overview of the effects of iron on primary production and associated carbon sequestration (Section 2.2.1), followed by an explicit description of our simulation with altered supply of bioavailable iron to the
ocean in Section 2.2.2.

## 2.1 Eddy parameterizations

### 2.1.1 Introduction

The simulation of oceanic motion (driven by pressure gradient force, gravity and viscous friction) by numerically solving the *primitive equations* is intimately coupled to the question of how to proceed with sub-grid processes that can not be resolved but
are known to affect processes on the resolved scales. The reason is that computational costs and constraints render it typically impossible to resolve all spatial scales that are involved in the dynamics of interest.

For global ocean-circulation models, that are coupled to biogeochemical (carbon) models, the explicit resolution of mesoscale processes has - so far - been already beyond computational capacities. Hence, to-date, model-based projections of oceanic carbon uptake rely on parameterizations of mesoscale processes.

Historically, attempts to parametrize mesoscale processes in ocean models started with relatively simple, horizontal, down-gradient *Laplacian* diffusion with associated constant diffusivities of the order of $1000\,\mathrm{m^2\,s^{-1}}$. Early on, it has been realised by Veronis (1975) that the associated horizontal mixing results in too intense diapycnal mixing in regions of sloped isopycnals (cf. McDougall and Church, 1985). As a workaround Redi (1982) proposed to transform the horizontal diffusion tensor such scalars and momentum are mixed only along isopycnals. The conundrum with this so-called *isopycnal mixing* scheme, however, is that
it (wrongly) implies that eddies do not have any effects on the dynamics in regions where the density distribution is governed by either temperature or salinity only.

To-date, virtually all climate models apply the parameterization of Gent and McWilliams (1990) (hereafter GM) to account for the effects of unresolved ocean eddies. GM constitutes a positive definite sink of the global potential energy by introducing a purely adiabatic extra advection. The parameterization is also referred to as *thickness diffusion* (even though its inventors now
consider the term to be misleading; Gent, 2011) which vividly describes the effect of thickness diffusion on layers bounded by two isopycnals – that is, evening out local differences in layer thickness. Associated to GM is a parameter, often dubbed *thickness diffusivity*, or $\kappa$, which prescribes the speed with which differences in isopycnal layer thicknesses are evened out. It





is agreed that $\kappa$ should be spatially varying in order to account for the fact that eddy-activity is not homogeneously distributed over the globe. But as concerns how it should or could be calculated in a coarse resolution model, there is no consensus.

The most pragmatic choice is setting $\kappa$ constant (cf. Gent et al., 1995). Other approaches include " ... an attempt to tune away model bias, rather than an attempt to make a poorly represented process more physical ..." (Gnanadesikan et al., 2005),

or are calibrated with results from eddy-resolving models (Eden and Greatbatch, 2008).

### 2.1.2 Sensitivity experiments - thickness diffusivities

The choice of the thickness diffusivity closure has been shown to cause local effects in the Southern Ocean such as differing Antarctic Circumpolar Current transports (Eden et al., 2009), and differing sensitivities of the meridional overturning circulation (MOC) towards wind stress changes (Viebahn, 2010). The question which closure yields the most realistic results has not

been unanimously answered yet because the rather strong bias in all of the simulations renders comparisons to observations inconclusive. As regards simulated MOC changes in response to changing winds, there is some guidance from intercomparison with eddy-resolving models (cf. Viebahn, 2010). This guidance, however, is based on the assumption that eddy-resolving models are realistic, albeit they are - as well - biased.

Our aim here is to quantify the uncertainty in the uptake of natural carbon in the Southern Ocean that is associated to the

choice of the closure for thickness diffusivity in a global coupled ocean circulation biogeochemical model. To this end, we compare results from three contemporary closures for the thickness diffusivity $\kappa$ dubbed *CON, FMCD, E&G*:

- CON; closure is constant in space and time: $\kappa = 600\,\mathrm{m^2\,s^{-1}}$.

- FMCD; closure is a function of space (longitude, latitude, tapering to the bottom and the surface) and time as a function of the horizontal density gradient averaged from 100 to 200 m depth:

$$\kappa = \alpha\overline{|\nabla_z\rho|}^z \left( \frac{L^2 g}{\rho_0 N_o} \right). \tag{1}$$

The dimensionless $\alpha = 0.07$, the length scale $L = 50\,\mathrm{km}$, and the Brunt-Väisälä frequency $N_0 = 0.004\,\mathrm{s^{-1}}$ are tuning constants. $g = 9.8\,\mathrm{m\,s^{-1}}$ is the standard acceleration of free fall. $\rho_0 = 1035$ is the reference density for the Boussinesq approximation and $\overline{|\nabla_z\rho|}^z$ is the average of the horizontal density gradient taken over the depth range 100 to 2000 m. Maximum and minimum values are set to 2000 and $200\,\mathrm{m^2\,s^{-1}}$, respectively.

- E&G; closure is a function of space (longitude, latitude, depth) and time as proposed by Eden and Greatbatch (2008) based on considerations of the eddy kinetic energy budget:

$$\kappa = L^2 \sigma, \tag{2}$$

where the inverse time scale $\sigma = |\partial_y\overline{b}|/N = |f\partial_z\overline{u}|/N$ is related to the Corioles parameter $f$, the vertical shear of the mean flow $\partial_z\overline{u}$ and the *Brunt-Väisälä* frequency $N$. $\sigma$ is also referred to as the Eady growth rate which is a measure of

the baroclinic instability. The length scale $L$ is set as $L = \min(L_R, L_{Rhi})$, with the local first baroclinic Rossby radius





$L_R$ and the Rhines Scale $L_{Rhi} = \sqrt{u/\beta}$. The Rhines Scale defines the spatial scale at which planetary rotation causes zonal jets. It is a function of the eddy horizontal velocity $u$ and $\beta$ (the latitudinal gradient of $f$).

In addition to the thickness diffusivities discussed above we apply an isopycnal diffusivity of $600\,\mathrm{m^2\,s^{-1}}$ in all configurations presented here.

## 2.2 Iron deposition

### 2.2.1 Introduction

Iron, although an abundant element on earth, is present only at very low concentrations in the ocean. Typically, the vertical distribution shows a profile which is similar to nitrate or phosphate (e.g. Johnson et al., 1997), with lower concentrations at the surface ($< 0.2\,\mathrm{nmol\,kg^{-1}}$) and concentrations peaking further down in the thermocline (at around $1\,\mathrm{nmol\,kg^{-1}}$). Another similarity to nutrients like nitrate or phosphate is that in vast oceanic regions, the growth of autotrophs is limited by the availability of iron (cf. Boyd and Ellwood, 2010). Among the iron-limited regions is the Southern Ocean where direct observational evidence shows that changes in iron supply affect the biotic uptake of carbon and its export to depth (Smetacek et al., 2012).

To this end a consensus has been reached in the literature. The global oceanic iron cycle is an important agent in the global biogeochemical carbon cycle. But, even so, there is still a large discrepancy between the evidential importance of iron dynamics and our poor quantitative understanding thereof. One expression of this discrepancy is that, on the one hand, the biogeochemical protocols for the CMIP6 Ocean Model Intercomparison Project (cf. Eyring et al., 2016) now rank simulated dissolved iron concentration as "priority 1" model output (Orr et al., 2016, their Table 5) while, on the other hand, the same protocols suggest not to initialize the models with observations of dissolved iron because suitable data compilations are not available yet.

In short, neither sources nor sinks of dissolved iron in the ocean are well constrained and data of standing stocks are so sketchy that the scientific community recommends not to use these data for simulations. Among the reasons for this dire situation are challenges such as (1) sources and sinks overlap spatially such that standing stocks of iron can not constrain the (speed of the) iron cycling (Frants et al., 2016), (2) aeolian sources of iron are intermittent and thus hard to quantify (e.g. Duggen et al., 2010; Olgun et al., 2011), (3) physico-chemical stabilization is not well understood such that e.g. the importance of iron originating from hydrothermal vents remains uncertain (Resing et al., 2015), and (4) iron sinks such as scavenging and precipitation are not well constrained (e.g. Tagliabue et al., 2014) and an explicit representation of the essential iron-binding ligand dynamics in models has only just begun (Völker and Tagliabue, 2015).

### 2.2.2 Sensitivity experiment - iron supply

The cycling of iron in the ocean is not well constrained: in terms of an average residence time, contemporary models differ by two orders of magnitudes (4 to 600 years Tagliabue et al., 2016). It is straightforward to assume that the large uncertainty in the supply and cycling of iron affects the sensitivity of simulated oceanic carbon uptake. As a first step towards relating uncertainties in iron dynamics with oceanic carbon uptake we conduct a sensitivity experiment were we change the aeolian supply





of bioavailable iron to the ocean. (Note that, to fathom the full range of uncertainty is beyond the scope of this manuscript.) The ratio behind this experiment is as follows: in a warming world the amount of dust air-borne is increasing due to vegetation loss, dune remobilization (e.g., Bhattachan et al., 2012) and glacier retreat (e.g., Bullard, 2013). In-line with this reasoning, model-aided estimates by Mahowald et al. (2010) suggest that the increase may well correspond to a doubling over the 20th

5 century over much of the globe. In our experimental design, we follow Krishnamurthy et al. (2009) and assume in our sensitivity experiment *IRON* that the deposition of bioavailable iron, associated to aeolian dust, does also double over a period of 50 years. Other than these changes of iron supply, the simulation IRON is identical to the simulation FMCD.

| Simulation Tag | Description |
| --- | --- |
| CON | The model configuration is identical to Galbraith et al. (2010) but with a constant thickness/eddy diffusivity of $600\,\mathrm{m^2\,s^{-1}}$. Starting from the semi-equilibrated state of Galbraith et al. (2010), we continue the spin-up by another 20 years, after which we increase the windspeed that drives the ocean south of $40^\circ$S. The respective increase is 14% in 50 years, consistent with results from a reanalysis of the period 1958 to 2007 (Lovenduski et al., 2013). |
| FMCD | The model configuration and integration procedure is identical to CON except for the eddy parameterisation which is set to the default CM2.1 setting in Galbraith et al. (2010) with a spatially varying thickness diffusivity. |
| E&G | The model configuration and integration procedure is identical to CON except for the eddy parameterisation which is set to spatially-varying following Eden and Greatbatch (2008) in its simplified form of Eden et al. (2009). |
| IRON | The underlying model configuration and integration procedure is identical to FMCD except for air-sea fluxes of iron south of $40^\circ$S which are increased by a rate corresponding to a doubling in 50 years. |

**Table 1.** Model simulations.





## 3 Results

Fig. 1 and Fig. 2 show thickness diffusivities in the configurations FMCD and E&G at the end of their spin-up: we find a $\approx 40\%$ difference in the average diffusivities between the contemporary concepts FMCD, E&G and the constant value of $600\,m^2\,s^{-1}$ in CON.

The thickness diffusivities in both FMCD and E&G vary substantially in space (Fig. 1) although not in unison: in FMCD we find large thickness diffusivities in the region of the Antarctic Circumpolar Current (ACC), which is in line with results from high resolution models and observations (e.g. Frenger et al., 2013; Hallberg and Gnanadesikan, 2006). In E&G the elevated values in the ACC are less pronounced and elevated diffusivities appear in the Weddell and Ross sea. In E&G, in contrast to FMCD, even the large values decay rapidly with depth (Fig. 2). Both FMCD and E&G peak locally at values twice as large

as the constant thickness diffusion applied in CON (Fig. 1). Even so the zonal averages of FMCD and E&G are typically lower than in CON (Fig. 2). The thickness diffusivities vary not only substantially among the configurations at the end of the respective spin-ups, but they do also feature differing sensitivities towards increasing winds. While the thickness diffusivities in CON stay constant at $600\,m^2\,s^{-1}$, the thickness diffusivities in both, FMCD and E&G, show a similar – albeit not identical – pattern of increase (Fig. 3a and b). Expressed in terms of a thickness diffusivity averaged over the whole Southern Ocean

the increase is linearly related to the wind increase and peaks at 16 and $25\,m^2\,s^{-1}$ in the configurations E&G and FMCD, respectively (Fig. 3c).

    In the following, we summarize the oceanic responses to the combination of increasing winds and changing thickness diffusivities:

– Ekman pumping. The vertical velocities that are driven by the horizontal divergence of Ekman transports increase along
with the winds by up to $50\,m\,yr^{-1}$ (Fig. 4). The responses of all considered configurations are almost identical (i.e. indistinguishable by eye), indicating that surface current/wind effects (cf. Dietze and Löptien, 2016) differ very little among the configurations. This also indicates that the kinetic energy transferred from the atmosphere to the ocean is very similar in all of the configurations.

– Meridional overturning. In all configurations the increased winds drive an enhanced meridional overturning with similar
patterns and amplitudes (Fig. 5). Also common to all configurations is that the fraction of meridional overturning that is effected by the repective GM parameterization is opposing the increasing trend (i.e. the changes in Fig. 6 are negative over most of the region). The magnitude of this counter effect, however, differs considerably among the configurations. Fig. 6 reveals that the counter effect, or "eddy compensation" as it is also referred to, peaks at 2 Sv in CON and E&G while FMCD features much higher values up to 4 Sv. Note that these results do not support the hypothesis that a more
complex definition of the thickness diffusivity (such as in E&G and FMCD) does necessarily amount to an increase of the (parameterized) eddy compensation relative to the original pragmatic choice (cf. Gent et al., 1995) of setting it constant (such as in CON).



– Air-sea heat flux. The air-sea heat flux averaged over the Southern Ocean is an indicator of diabatic heatfluxes in its thermocline. After all heat exchanged between the surface mixed layer and the atmosphere has to be replaced from somewhere. (This view neglects trends in sea surface temperature and near-surface oceanic heat fluxes entering the region). Fig. 7a shows the temporal evolution of heat exchanged with the atmosphere. The most striking feature is that
in all configurations the air-sea heat exchange is relatively constant, even though the winds increase considerably. Other than that, there is a small offset between the configurations, indicating differences in the meridional overturning of heat. Short term anomalies in time, such as between year 30 and 40 in FMCD and E&G in Fig. 7a, are correlated with sea-ice extent.

– Sea ice cover. Among the biggest concern in preparation of the configurations was sea ice. The reason is that sea ice
caps the ocean shielding it from air-sea exchange of heat and carbon. Thus, sea-ice dynamics is coupled to carbon uptake and a comparison of model configurations that feature differing sea-ice covers can be challenging. The intimate coupling between sea-ice cover and air-sea fluxes is evident in Fig. 7a and b. All anomalies in the heat fluxes (such as e.g. between year 30 and 40 in E&G and FMCD) have their counterpart in sea ice extent, suggesting that less sea-ice results in more cooling of the ocean as vaster areas of relatively warm ocean waters are exposed to the cold polar atmosphere.
Fortunately, the temporal evolution of sea ice extent is very similar among the configurations. If it were not (such as e.g. in the configurations compared by Bryan et al., 2014), the interpretation of results would not be straightforward.

Despite some significant differences in simulated physics among the configurations FMCD, E&G and CON (such as, e.g., differing levels of eddy-compensation) the simulated differences in oceanic carbon uptake are small (Fig. 7c). So small that we conclude that the simulated oceanic carbon uptake in the Southern Ocean is rather robust towards the choice of details in the
contemporary eddy-paramterization of Gent and McWilliams (1990). In each of our configurations FMCD, E&G and CON the carbon uptake of the Southern Ocean decreases almost linearly in time. Eventually, the Southern Ocean turns from a sink to a source of natural carbon, irrespective of the eddy-parameterization. Other than that there are small deviations from a linear decrease such as e.g. the "anomaly" between year 60 and 70 in configuration CON. These deviations correspond to changes in ice extent with less sea ice being associated with stronger outgassing (or less uptake) of carbon.
A similarly robust behaviour is, however, not inherent to biogeochemical module. By changing the deposition of bioavailable iron within its "envelope of uncertainty" in experiment IRON the oceanic carbon uptake does change substantially. Fig. 7 shows that even the sign of air-sea carbon fluxes in the Southern Ocean changes relative to all the other simulations.

## 4   Summary and conclusion

Global coupled ocean-circulation biogeochemical models predict an increase of oceanic natural $CO_2$-outgassing due to strength-
ening winds in the Southern Ocean (e.g. Lovenduski et al., 2013). These predictions contain a considerable degree of uncertainty, some of which that is associated to what Lovenduski et al. (2016) refer to as inter-model "structural differences".

In the present study, we compare two sources of uncertainties in simulated carbon uptake in response to increasing winds in the Southern Ocean with one another: (1) The uncertainty related to actively discussed details in the contemporary (GM-)





paramterization of Gent and McWilliams (1990) which mimics the effects of unresolved mesoscale circulation on the resolved larger scale circulation in coarse resolution models. Specifically, we explore different definitions of the respective thickness diffusivity. (2) To put the results into perspective, we also consider the uncertainty that is related to the rather unconstrained deposition of bioavailable iron to the sun-lit surface ocean.

The investigation of the GM-parametrization is motivated by studies such as of Farnetti and Gent (2011); Gent and Danabasoglu (2011), who argue that a variable, rather than a constant, thickness diffusivity is key to a realistic effect of unresolved mesoscale physical processes on the resolved (coarse resolution) circulation that - in turn - is an essential precondition for a realistic response of the Southern Ocean to stronger winds. Lovenduski et al. (2013) support this view and find, indeed, that the sea-air $CO_2$-outgassing is damped by (parameterized) eddy compensation in scenarios with strengthening winds. To

this end, our results based on a suite of model simulations are consistent in that they also show an eddy compensation that significantly dampens the increase of the meridional overturning circulation in response to increasing winds. As regards the magnitude of the response on air-sea carbon fluxes we find, however, that differences between contemporary approaches to define the thickness diffusivity of Gent and McWilliams (1990) are small. Or, in other words, GM's eddy-parameterization is relatively robust towards the choice of the respective scaling coefficient (i.e. the thickness diffusivity). In our opinion this

enhances the credibility of GM's seminal parameterization. This is fortunate because a high sensitivity towards the choice a scaling coefficient would not be a good base for a projection of oceanic carbon uptake in a warming world.

In contrast, our results indicate that the biogeochemical module tested here, does not yet feature a robust response. Specifically we explored the uncertainty that is associated with the air-sea deposition of bioavailable iron which - on its own - prevents the specification of even the sign of air-sea carbon fluxes in a world of increasing winds. Note that the overall uncertainty due

to the biogeochemical component must be much higher as not only the iron supply tested here is uncertain, but also, the residence time of iron varies by two order of magnitudes among contemporary biogeochemical models (Tagliabue et al., 2016). Additional uncertainty is associated with the Michaelis-Menten formulation - a concept which is generic to biogeochemical modelling and which describes the limitation of autotrophic growth whenever essential resources (such as iron) are depleted. Although the Michaelis-Menten formulation is generic it is discussed controversially. Developed with enzyme kinetics in mind

it may not be applicable to autotrophs (e.g., Smith et al., 2009), and respective parameters (so-called half saturation constants) may be impossible to constrain with typical observations (Löptien and Dietze, 2015), even though they exert crucial control on the models' solutions.

In summary, our results indicate that a poor quantitative understanding of biogeochemical processes is a major source for uncertainties in model-based estimates of oceanic uptake of natural carbon in the Southern Ocean. Ranked against uncertainties

associated with the choice of the thickness diffusivity, the impression is that uncertainties in biogeochemical processes dominate. Given that the biogeochemical modules, in contrast to the physical modules, are not build based on first principles (such as Newton's Laws) this may not be astounding. Note, however, that there are regions where the opposite hold: e.g. findings by Dietze and Löptien (2012) suggest that an incomprehensive understanding of physical rather than biogeochemical processes prevents a realistic modelling of biogeochemical processes in the thermocline of the eastern tropical Pacific.



A caveat remains. Although we showed that GM's parameterization is, in terms of the carbon uptake in the Southern Ocean, rather robust - still - all of our coarse-resolution simulations could be biased. To this end, the increase in computer power is about to provide some guidance now that recent configurations can afford to resolve much of the mesoscale (e.g. Dufour et al., 2015; Bishop et al., 2016) in the Southern Ocean - albeit they have no explicit representation of the carbon cycle yet. Hence a comparison of the sensitivity of carbon uptake to increasing winds between coarse resolution models (like the configurations tested here) and configurations that explicitly resolve mesoscale processes, is to come. For the time being Gent (2016) summarizes the respective field of physics-only configurations by stating that high-resolution models have approximately 50% compensation of the MOC-increase. Thus, all the simulations shown here may underestimate the eddy compensation by a factor of two.

**Appendix A: Model assessment**

A comprehensive evaluation of the spun-up configuration FMCD is provided by Galbraith et al. (2010) who show that the model is competitive in the sense that its deviations from observations is similar to what can be expected from the current generation of earth system models. Here, we show a small choice of model-data comparisons only:

– Sea surface temperature (SST, Fig. 8) is associated to oceanic carbon storage via the solubility of water that is in contact with the atmosphere. Further, SST biases are indicative of deficient physics in setups like ours, where SST is not restored but effected by the entanglement of air-sea heat fluxes with ocean circulation.

– Surface phosphate concentrations (Fig. 9) are indicative for the efficiency of the biological carbon pump to draw down surface nutrients which are continuously re-supplied to the sun-lit surface by upwelling and vertical diffusive processes.

– Zonally averaged meridional section of oxygen concentrations (Fig. 10) are indicative for the balance between deep ocean ventilation and biotic oxygen consumption.

– Sea ice cover (Fig. 11) is supposedly, according to e.g. Bryan et al. (2014), key to simulated air-sea carbon fluxes as it can cap the air-sea exchange of gases. Note that all of our simulations feature very similar ice cover (which is not shown explicitly here, but indicated by the similar ice extent in Fig. 7 c).

Closer inspection of Fig. 8 to 11 reveals typical model deficiencies, among them (1) a SST cold bias (Fig. 8) in the eastern tropical Pacific and Indic, (2) a spurious nutrient drawdown in oligotrophic regions (Fig. 9) which may be the consequence of a wide-spread, but flawed, phytoplankton growth concept (Smith et al., 2009), (3) an equatorial oxygen deficit (Fig. 10) which is related to unresolved physics (Dietze and Löptien, 2012; Getzlaff and Dietze, 2013), (4) an oxygen distribution which is biased high polewards of $60°$N and close to Antarctica which is probably associated with deficient deep water formation, (5) overestimated sea ice melting in Summer around Antarctica (as can be derived from Fig. 11).

Here we conclude that all of our semi-equilibrated model configurations (listed in Tab. 1) feature simulations that deviate by roughly equal amounts from the observations. This, in its turn, suggests that the simulated sensitivities of any of our



configurations towards changing winds in the Southern Ocean, is equally likely. Or in other words, none of our simulations can be discarded nor favoured with the argument that the respective simulated mean state is especially unrealistic or realistic.

*Author contributions.* All authors were involved in the design of the work, in data analysis, in data interpretation and in drafting the article.

*Acknowledgements.* Eric Galbraith, contributor to the MOM (www.gfdl.noaa.gov/mom-ocean-model/) community and developer of BLING
5  (www.sites.google.com/site/blingmodel/), shared his model configuration with us. We are grateful to him and the rest of the MOM community! All authors acknowledge long-term support by Andreas Oschlies. J. G. acknowledges funding by *Deutsche Forschungsgemeinschaft* via the project "Impact of eddy parameterisations on the simulated response of Southern Ocean air-sea $CO_2$-fluxes to wind stress changes in IPCC-type ocean models". Integrations were performed on the compute clusters weil.geomar.de, wafa.geomar.de, and other hardware from the GEOMAR Helmholtz Centre for Ocean Research, Germany, Kiel, FB2/BM. Further, we used the scalar HPC cluster of the NEC System
10 at the Christian-Albrechts-Universität zu Kiel which is co-funded by GEOMAR. The authors wish to acknowledge use of the Ferret program for analysis and graphics in this paper. Ferret is a product of NOAA's Pacific Marine Environmental Laboratory. (Information is available at http://ferret.pmel.noaa.gov/Ferret/)



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




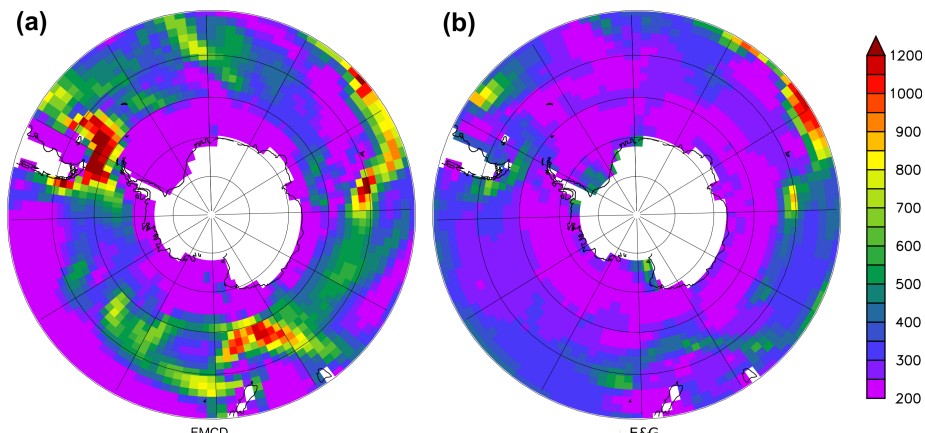

**Figure 1.** Vertical mean thickness diffusivities (upper 500 m), averaged over the last 20 years of spin-up. Subpanel (a) refers to configuration FMCD and (b) refers to configuration E&G. The units are $m^2 s^{-1}$.

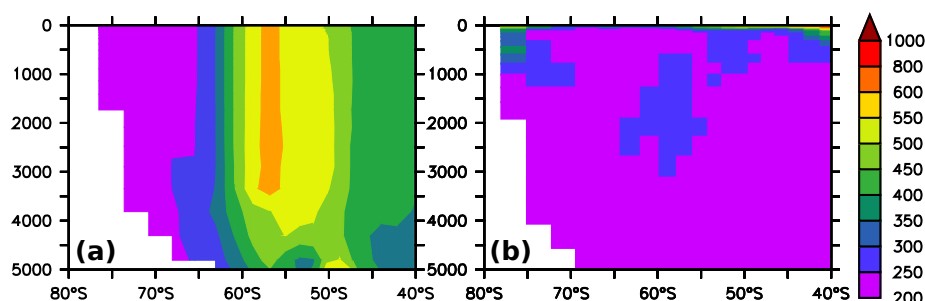

**Figure 2.** Zonal mean thickness diffusivities, averaged over the last 20 years of spin-up. Subpanels (a) and (b) refer to configuration FMCD and E&G, respectively. The units are $m^2 s^{-1}$.





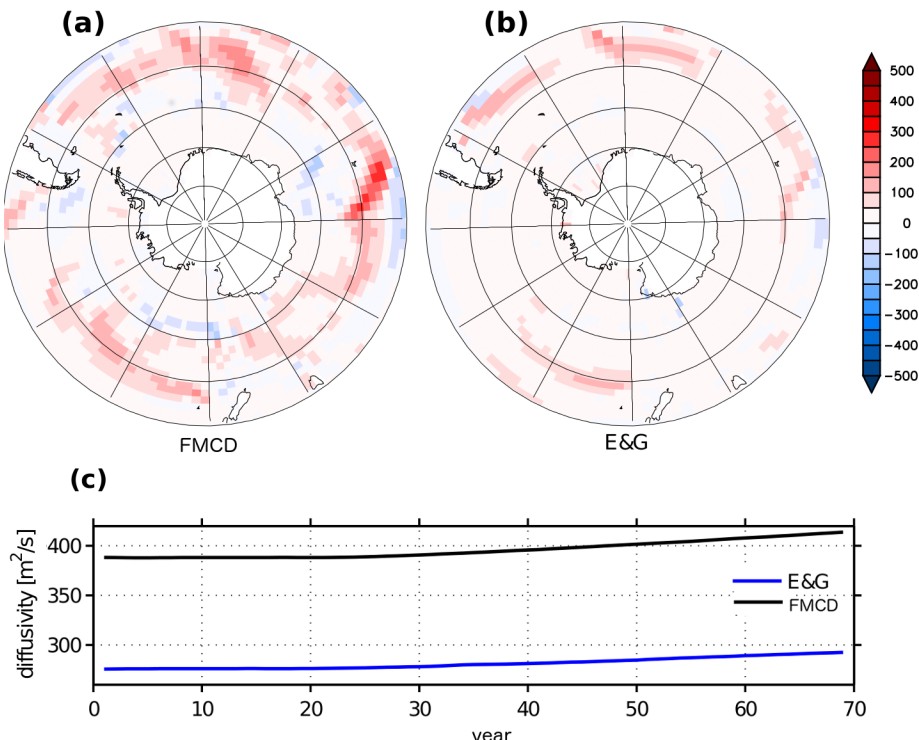

**Figure 3.** Temporal evolution of thickness diffusivity, average over the upper 1000 m. (a) and (b) refer to changes (the average of year 45-49 minus the average over the last 20 years of the spin-up) effected by increasing winds in simulation FMCD and E&G, respectively. (c) shows the evolution of the domain-averaged (south of $40°$S) diffusivity simulated with FMCD (black line) and E&G (blue line). The first 20 years correspond to the end of respective spin-ups. From year 20 onward, the winds increase. The units are $m^2 s^{-1}$. Configuration CON (with a constant $600 \, m^2 \, s^{-1}$) is not shown.





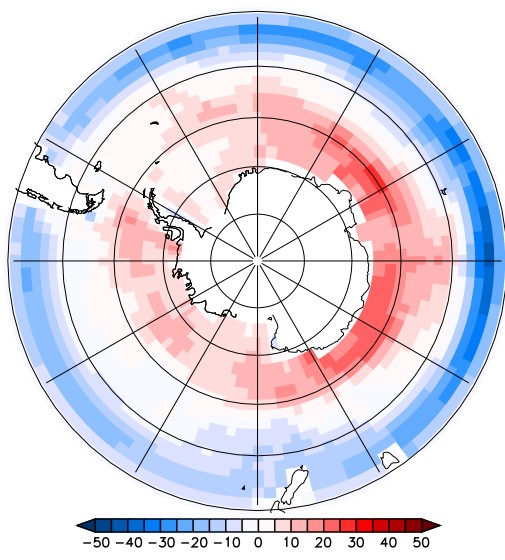

**Figure 4.** Acceleration of Ekman pumping as simulated with configuration FMCD (and IRON). The unit is $\mathrm{m\,yr^{-1}}$. Regions with increased upwelling (or reduced pumping) are coloured in red.

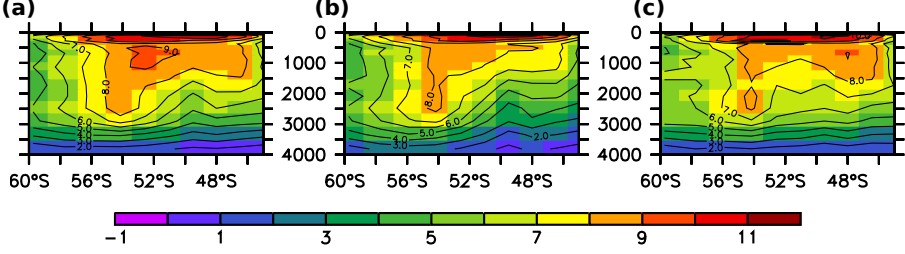

**Figure 5.** Change in meridional overturning (after 49 years of increasing winds) in units Sv ($10^6\ \mathrm{m^3\,s^{-1}}$). Positive values indicate increasing overturning in response to increasing winds. (a), (b) and (c) refer to simulations CON, FMCD and E&G, respectively.



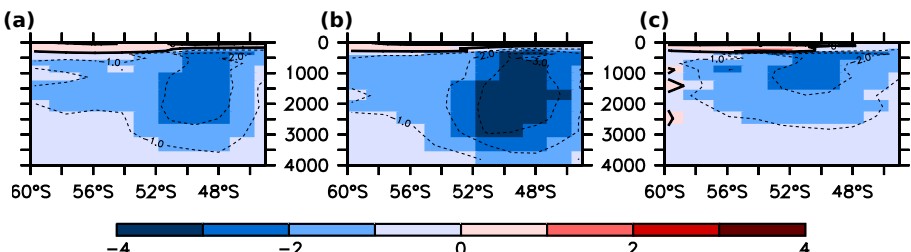

**Figure 6.** Change of that fraction of the meridional overturning that is effected by the respective GM parameterizations (after 49 years of increasing winds) in units Sv ($10^6 \, \mathrm{m}^3 \, \mathrm{s}^{-1}$). Negative values indicate a damping of the overturning in response to increasing winds. (a), (b) and (c) refer to simulations CON, FMCD and E&G, respectively.

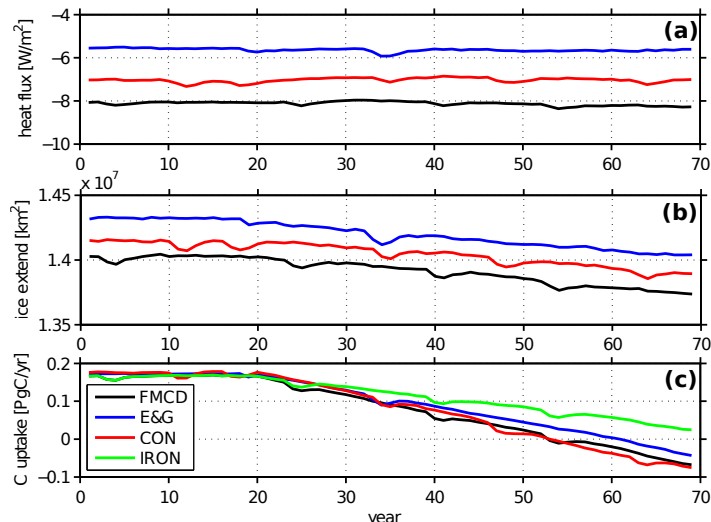

**Figure 7.** Oceanic response to increasing winds south of 40°S (spatially and annually averaged). (a) shows air-sea heat fluxes with negative values denoting oceanic cooling), (b) shows ice-covered area and (c) shows oceanic carbon uptake with positive values denoting oceanic uptake. The line colors refer to experiments as indicated in the legend. The first 20 years correspond to the end of respective spin-ups. From year 20 onward, the winds increase.



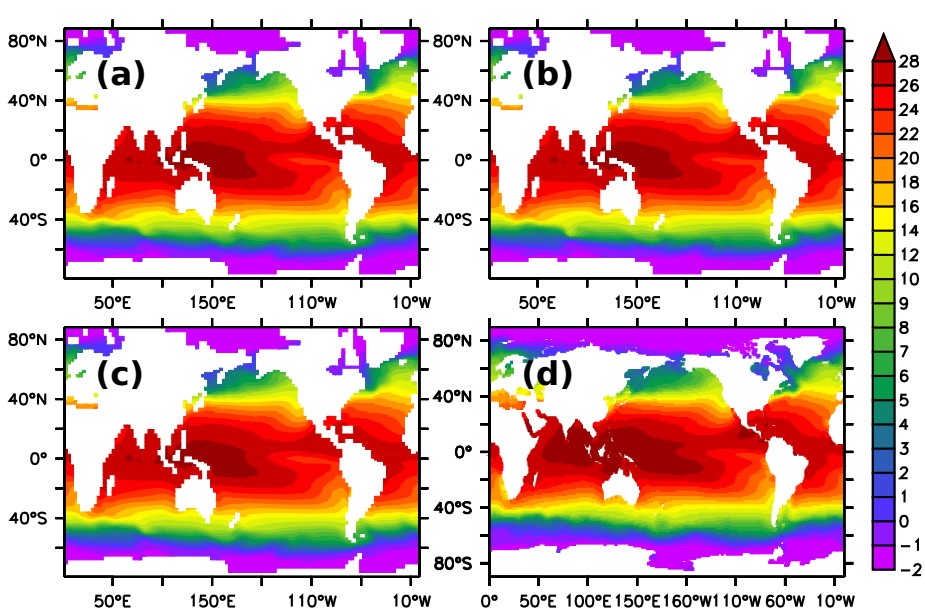

**Figure 8.** Annual mean sea surface temperature in units °C. **(a)**, **(b)** and **(c)** refer to spun-up states of configurations FMCD, CON and E&G, respectively. **(d)** shows observations (Locarnini et al., 2010).





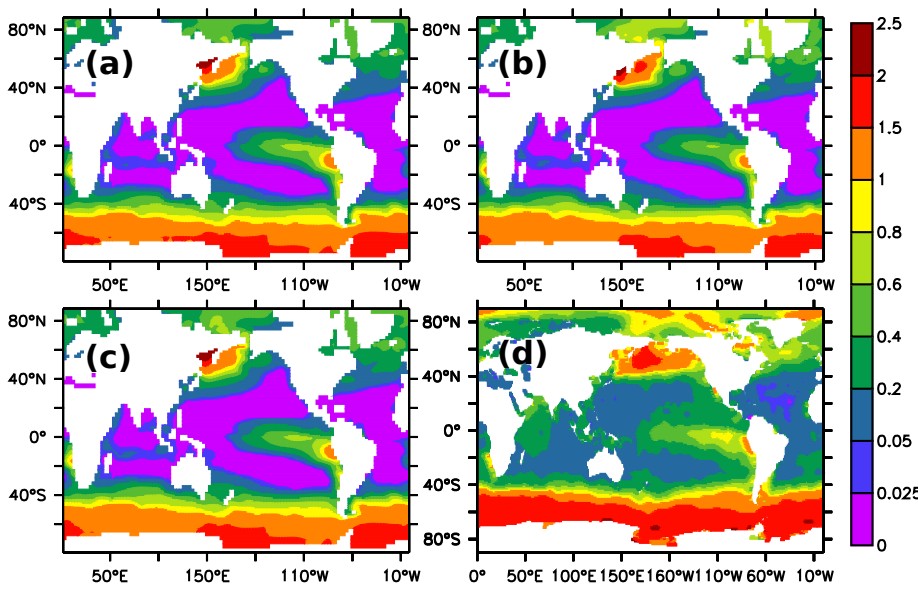

**Figure 9.** Annual mean surface phosphate concentration in units $\mathrm{mmol\,P\,m^{-3}}$. The colour scale is nonlinear and highlights low, limiting concentrations. **(a)**, **(b)** and **(c)** refer to simulations FMCD, CON and E&G, respectively. **(d)** shows observations (Garcia et al., 2010a).





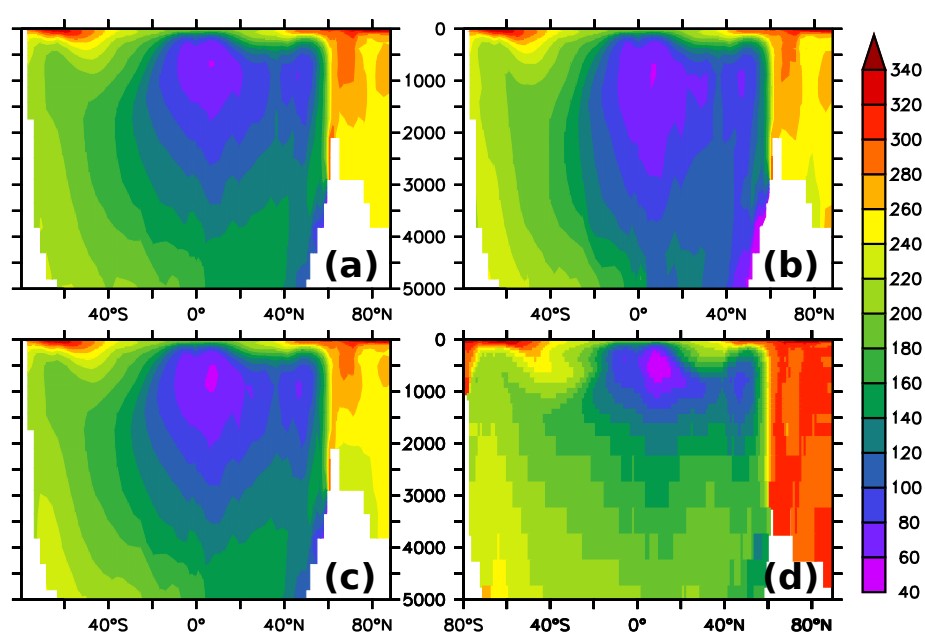

**Figure 10.** Meridional section of annual mean oxygen concentration in units $mmol\,O_2\,m^{-3}$ (zonally averaged). **(a), (b)** and **(c)** refer to simulations FMCD, CON and E&G, respectively. **(d)** shows observations (Garcia et al., 2010b).



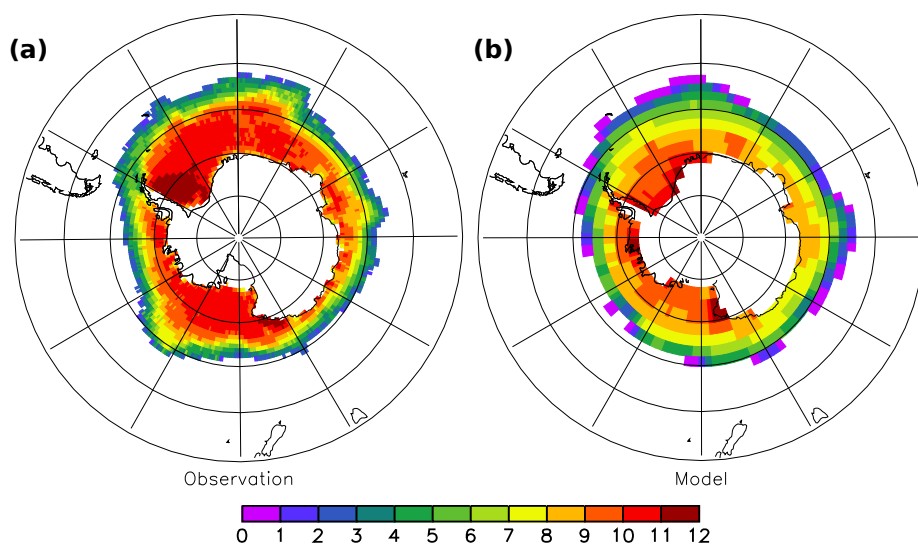

**Figure 11.** Ice covered months in a year. **(a)** refers to a 1990 to 2000 average derived from the Rayner et al. (2003) global analysis. **(b)** refers to the simulation FMCD.