# Peer review of "Simulating natural carbon sequestration in the Southern Ocean: on uncertainties associated with eddy parameterizations and iron deposition"

_Biogeosciences, 2016_

## Referee Comment (RC1) · P. R. Gent (Referee) · 22 Nov 2016

This manuscript should definitely be published, but I have several comments that need to be addressed.

1) Table 1: In most climate model experiments where the zonal wind stress has been increased, the increased wind speed has not been applied to the heat and fresh water flux terms. I suspect this is also the case for these experiments because the air-sea heat exchange is described as relatively constant (Pg 9, l 5). This definitely needs to be clarified and stated.

2) Pg 8, l 30-32. A constant GM coefficient can only produce marginal eddy compensation (Fig 6a). A variable GM coefficient is required to produce significant eddy compensation, but some choices do not (Fig 6c).

3) Fig 7c shows different rates of decline in oceanic carbon uptake in the four different experiments performed. I think the linear slopes over years 20-70 should be calculated and compared. This will produce some change between the E&G (blue) slope and the CON and FMCD slopes that is about 20% as large as the slope change in the IRON (green) slope. Is a 20% change "rather robust" as described on pg 9 l 19? It is also unfair to the IRON simulation to say it has the wrong sign of air-sea carbon fluxes (pg 9 l 27), because if the experiment were extended another 10 years, then the sign of the IRON curve in Fig 7c would almost certainly be negative. A better comparison would be the linear slope values. Should spatial maps of the oceanic carbon uptake changes be shown?

4) Pg 11, l 1. A caveat of the present results is that the horizontal resolution of the ocean model is very coarse at 3 deg. Most climate models use a resolution of 1 deg or finer. At NCAR, we now rarely use our 3 deg ocean model because it just doesn't have enough resolution to represent several aspects of the ocean circulation, including the Southern Ocean. I would like to see a comparison like this using 1 deg resolution ocean models to see whether the present conclusions hold, because comparisons with 0.1 deg ocean models with biogeochemistry are still a few years away.

5) Figs 8-10. I would prefer to see observations and then the model minus observations differences, especially in the SSTs in Fig 8.

Pg 12, l 2. I disagree. Figs 1, 3 and 5 clearly show that the FMCD choice has a better spatial representation of eddy kinetic energy compared to observations. It also shows a much stronger eddy compensation, which is more in line with eddy-resolving model results. I think it looks a much better choice than E&G or a constant: it really is about time to go beyond using a constant GM coefficient in global climate models.

Minor Comments:

1) Pg 1, l 21. The changes in the Southern Hemisphere atmosphere have been driven by changes in the ozone hole as well as by greenhouse gases: Polvani et al (2011), J. Climate, 24, 795.

2) Pg 2, l 7. There is also recent evidence that the Southern Ocean carbon sink has been "reinvigorated": Landschutzer et al (2015), Science, 349, 1221.

3) Pg 5, l 10-12. There aren't observations of the Southern Ocean MOC, and Bryan et al (2014) should also be referenced here.

4) Pg 5, l 28. Coriolis.

5) Pg 7, l 2. Rationale.

6) Pg 8, l 26. Respective.

7) Pg 10, l 8. Reference Swart et al (2014), Biogeosciences, 11, 6107.

————————————————

---

## Referee Comment (RC2) · Anonymous Referee #2 · 2 Jan 2017

GENERAL COMMENTS:

This manuscript describes differences in Southern Ocean (south of 40S) carbon uptake for several coarse resolution (3 degree) model simulations with different eddy-parameterizations and a simulation with increased bioavailable iron. In all simulations, winds are increased by approximately 15% over 50 years following spin-up. The major result is that the decrease in Southern Ocean carbon uptake due to increased winds is not significantly different in the three simulations with very different eddy-parameterizations. In contrast, the simulation with increased bioavailable iron shows

a markedly smaller decrease in carbon uptake over 50 years relative to the case with constant bioavailable iron. Other recent studies have suggested that uncertainties in ocean biogeochemistry are of lesser importance with respect to future Southern Ocean carbon uptake compared to uncertainties related to eddy-parameterizations.

This manuscript is concise and focused with an important conclusion and should be published after minor revisions.

SPECIFIC COMMENTS:

Pg. 2, Lines 5-7: I would cite more recent studies here and include recent analyses of observations. Up to the mid 2000s there is evidence from models (e.g., Le Quéré et al., 2007; Lovenduski et al., 2007) and observations (please cite Landschützer et al., 2015) that Southern Ocean carbon uptake may have slowed relative to the expected increase due to the increase in atmospheric CO2. More recent observational studies (please cite Landschützer et al., 2015; Munro et al., 2015; and Xue et al., 2015) suggest that the sink may have strengthened over the last decade.

Pg. 2, Line 8: I would say something more general like the "the link between variability in surface winds and Southern Ocean carbon uptake remains inconclusive"

Pg. 2, Lines 16-22: I would also mention current observational/model studies that have examined carbon uptake associated with mesoscale eddies within the Southern Ocean (please cite Song et al., 2016). This paper includes an analysis of the Drake Passage Time-series which represents the densest dataset of pCO2 observations within the ACC. Observations are compared to results from a high-resolution (approximately 0.1 degree) simulation of the Southern Ocean region surrounding the Drake Passage. Both observations and model output indicate how a shifting balance of physical and biogeochemical processes drive air-sea carbon flux during different seasons and gives important context to the complexity of the topic presented here.

Figures: Fig. 7 is the most important in the paper particularly Fig. 7c. I think it would

be helpful to include a Table summarizing these results with the linear rate of decrease in C uptake with uncertainty over the 50 years of increased winds. Alternatively, you could present the difference in C uptake with uncertainty between the last twenty years of spin-up and the last five or ten years of increase winds (i.e., years 46-50 or 41-50).

Figs. 8-11 might be more appropriate in a supplemental information section if allowed so that the reader focuses on the figures most important to the overall story.

TECHNICAL COMMENTS:

Pg. 2, Line 2: Replace "as regards their" with " with regards to their"

Pg. 3, Line 11: Replace "subsections 2.2.1 and 2.2.1" with "subsections 2.1.2 and 2.2.2"

Pg. 4, Line 3: Replace "following sub-section" with "following subsections"

Pg. 6, Line 32: Replace "were" with "where"

Pg. 7, Line 2: Replace "ratio" with "rational"

Pg. 9, Line 2: Delete "." Replace "After all" with "since"

Pg. 9, Line 8: Delete "."

Pg. 9, Line 8: Don't capitalize "T" in first word within parenthesis or remove parenthesis

Pg. 9, Line 10: Replace "is" with "are"

Pg. 9, Line 15: Replace "such as e.g." with "e.g., as"

Pg. 9, Line 19: Replace "towards the choice of" with "with respect to"

Pg. 11, Line 14: Replace "to" with "with"

Pg. 11, Line 17: Replace "indicative for" with "indicative of"

Pg. 11, Line 25: Replace "Indic" with "Indian Ocean"

Pg. 12, Line 8: Replace "compute" with "computer"

REFERENCES:

Landschützer, P., et al. (2015), The reinvigoration of the Southern Ocean carbon sink, Science 349(6253), 1221 – 1224, doi:10.1126/science.aab2620.

Munro, D.R., N.S. Lovenduski, T. Takahashi, B.B. Stephens, T. Newberger, and C. Sweeney (2015), Recent evidence for a strengthening CO2 sink in the Southern Ocean from carbonate system measurements in the Drake Passage (2002–2015), Geophys. Res. Lett. 42(18), 7623 – 7630, doi:10.1002/2015GL065194.

Song, H., J. Marshall, D.R. Munro, S. Dutkiewicz, C. Sweeney, D.J. McGillicuddy Jr., and U. Hausmann (2016), Mesoscale modulation of air-sea CO2 flux in Drake Passage, J. Geophys. Res. 121(8), 6635, doi:10.1002/2016JC011714.

Xue, L., L. Gao, W.-J. Cai, W. Yu, and M. Wei, 2015. Response of sea surface fugacity of CO2 to the SAM shift south of Tasmania: Regional differences, Geophys. Res. Lett. 42, 3973 – 3979, doi:10.1002/2015GL063926.

---

## Author Comment (AC1) · 25 Jan 2017

Review #1; RC1

Dear Peter R. Gent,

thank you for your time, work and very constructive comments!
Below, please find our responses to the issues you have raised.

Yours sincerely,
the authors

**Point-by-point responses:**

**1) Table 1: In most climate model experiments where the zonal wind stress has been increased, the increased wind speed has not been applied to the heat and fresh water flux terms. I suspect this is also the case for these experiments because the air-sea heat exchange is described as relatively constant (Pg 9, l 5). This definitely needs to be clarified and stated.**

**-A:** Agreed - this indeed needs clarification in the revised manuscript. We applied the increased winds to all bulk formulas including those for heat and freshwater.

**2) Pg 8, l 30-32. A constant GM coefficient can only produce marginal eddy compensation (Fig 6a). A variable GM coefficient is required to produce significant eddy compensation, but some choices do not (Fig 6c).**

**-A:** This makes sense to us - we will change the text in the revised version of the manuscript accordingly.

**3) Fig 7c shows different rates of decline in oceanic carbon uptake in the four different experiments performed. I think the linear slopes over years 20-70 should be calculated and compared. This will produce some change between the E&G (blue) slope and the CON and FMCD slopes that is about 20% as large as the slope change in the IRON (green) slope. Is a 20% change "rather robust" as described on pg 9 l 19? It is also unfair to the IRON simulation to say it has the wrong sign of air-sea carbon fluxes (pg 9 l 27), because if the experiment were extended another 10 years, then the sign of the IRON curve in Fig 7c would almost certainly be negative. A better comparison would be the linear slope values. Should spatial maps of the oceanic carbon uptake changes be shown?**

**-A:** We will add a discussion concerning linear slopes to the revised version of the manuscript. This will make the interpretation of our results less vague (and more robust).

We are still undecided as concerns the presentation of spatial maps of the oceanic carbon uptake changes because they look so similar (please see below).

[Figure]

*Simulated trends of air-sea carbon exchange associated to the linear increase in wind speed. The units are mmolC/m$^2$/yr$^2$. Positive values denote increasing (decreasing) oceanic outgassing (uptake). (a) refers to simulation FMCD, (b) to simulation CON, (c) to simulation E&G, and (d) to simulation IRON.*

**4) Pg 11, l 1. A caveat of the present results is that the horizontal resolution of the ocean model is very coarse at 3 deg. Most climate models use a resolution of 1 deg or finer. At NCAR, we now rarely use**

**our 3 deg ocean model because it just doesn't have enough resolution to represent several aspects of the ocean circulation, including the Southern Ocean. I would like to see a comparison like this using 1 deg resolution ocean models to see whether the present conclusions hold, because comparisons with 0.1 deg ocean models with biogeochemistry are still a few years away.**

**-A:** We agree that there is a caveat and will add the respective information (and citation) in the revised version of the manuscript. As concerns the comparison with higher-resolution models: we are currently working on a 0.1 deg configuration with full biogeochemistry: c.f. http://89.27.255.63/?page_id=90 and https://www.youtube.com/channel/UCnuABRT7qWGgM6bvMzLpr6A and we hope that we can present the respective comparison soon in an additional publication.

**5) Figs 8-10. I would prefer to see observations and then the model minus observations differences, especially in the SSTs in Fig 8.**

**-A:** We will show observed SSTs and model minus observations in the revised version of the manuscript.

**Pg 12, l 2. I disagree. Figs 1, 3 and 5 clearly show that the FMCD choice has a better spatial representation of eddy kinetic energy compared to observations. It also shows a much stronger eddy compensation, which is more in line with eddy-resolving model results. I think it looks a much better choice than E&G or a constant: it really is about time to go beyond using a constant GM coefficient in global climate models.**

**-A:** O.K. We pushed too far in the appendix. In the revised version of the manuscript we will remove the sentence " ... This, in its turn, suggests that the simulated sensitivities of any of our configurations towards changes in the Southern Ocean, are equally likely".

**1) Pg 1, l 21. The changes in the Southern Hemisphere atmosphere have been driven by changes in the ozone hole as well as by greenhouse gases: Polvani et al (2011), J. Climate, 24, 795.**

**-A:** We will add the respective information (and citation) to the revised version of the manuscript.

**2) Pg 2, l 7. There is also recent evidence that the Southern Ocean carbon sink has been "reinvigorated": Landschutzer et al (2015), Science, 349, 1221.**

**-A:** We will add the respective information (and citation) to the revised version of the manuscript.

**3) Pg 5, l 10-12. There aren't observations of the Southern Ocean MOC, and Bryan et al (2014) should also be referenced here.**

**-A:** We will add the respective information (and citation) to the revised version of the manuscript.

**4) Pg 5, l 28. Coriolis.**

**-A:** O.K.

**5) Pg 7, l 2. Rationale.**

**-A:** O.K.

**6) Pg 8, l 26. Respective.**

**-A:** O.K.

**7) Pg 10, l 8. Reference Swart et al (2014), Biogeosciences, 11, 6107.**

**-A:** Agreed! We will add this reference to the revised version of the manuscript!

---

## Author Comment (AC2) · 25 Jan 2017

Review #2; RC2

Dear anonymous referee #2,

thank you for your work! As you will see in the point-by-point responses below, all of your remarks make sense to us. We are especially grateful for pointing us to additional recent literature in such a constructive way.

Yours sincerely,
the authors

**Point-by-point responses:**

**Pg. 2, Lines 5-7: I would cite more recent studies here and include recent analyses of observations. Up to the mid 2000s there is evidence from models (e.g., Le Quéré et al., 2007; Lovenduski et al., 2007) and observations (please cite Landschützer et al., 2015) that Southern Ocean carbon uptake may have slowed relative to the expected increase due to the increase in atmospheric CO2. More recent observational studies (please cite Landschützer et al., 2015; Munro et al., 2015; and Xue et al., 2015) suggest that the sink may have strengthened over the last decade**.

**-A: Thanks** - we will add the respective information/references to the revised version of the manuscript.

**Pg. 2, Line 8: I would say something more general like the "the link between variability in surface winds and Southern Ocean carbon uptake remains inconclusive"**

**-A:** Agreed - will be changed in the revised version of the manuscript.

**Pg. 2, Lines 16-22: I would also mention current observational/model studies that have examined carbon uptake associated with mesoscale eddies within the Southern Ocean (please cite Song et al., 2016). This paper includes an analysis of the Drake Passage Timeseries which represents the densest dataset of pCO2 observations within the ACC. Observations are compared to results from a high-resolution (approximately 0.1 degree) simulation of the Southern Ocean region surrounding the Drake Passage. Both observations and model output indicate how a shifting balance of physical and biogeochemical processes drive air-sea carbon flux during different seasons and gives important context to the complexity of the topic presented here.**

**-A:** Agreed - we will add the respective information/references to the revised version of the manuscript.

**Figures: Fig. 7 is the most important in the paper particularly Fig. 7c. I think it would be helpful to include a Table summarizing these results with the linear rate of decrease in C uptake with uncertainty over the 50 years of**

**increased winds. Alternatively, you could present the difference in C uptake with uncertainty between the last twenty years of spin-up and the last five or ten years of increase winds (i.e., years 46-50 or 41-50).**

**-A:** This is in-line with recommendations/suggestions from the other reviewer and makes sense to us. We will add a table with the respective information to the manuscript.

**Figs. 8-11 might be more appropriate in a supplemental information section if allowed so that the reader focuses on the figures most important to the overall story.**

**-A:** We will explore this option.

**TECHNICAL COMMENTS:**
**Pg. 2, Line 2 ...**

**-A:** We will apply all your corrections in the revised version of the manuscript. Thank you for combing through so thoroughly.

---

## Author Response (AR1)

Kiel, 28th February 2017

Dear Editor,

Please consider our revised manuscript "Simulating natural carbon sequestration in the Southern Ocean: on uncertainties associated with eddy parameterizations and iron deposition" for publication in Biogeosciences.

The reviewers comments were extraordinarily constructive! They resulted in changes to the text marked in bold. We elaborate on this point-by-point below.

Thank you (and the reviewers) for your (their) time and work!

Yours sincerely,

the authors
* * *
* * *
**Point-by-point - corrections triggered by Review #1; RC1**

**1) Table 1: In most climate model experiments where the zonal wind stress has been increased, the increased wind speed has not been applied to the heat and fresh water flux terms. I suspect this is also the case for these experiments because the air-sea heat exchange is described as relatively constant (Pg 9, l 5). This definitely needs to be clarified and stated.**

**-A:** We applied the increased winds to all bulk formulas including those for heat and freshwater. Clarified on pg.4, ln.4.

**2) Pg 8, l 30-32. A constant GM coefficient can only produce marginal eddy compensation (Fig 6a). A variable GM coefficient is required to produce significant eddy compensation, but some choices do not (Fig 6c).**

**-A:** Our impression (although we have no proof) is that a constant but high GM coefficient would already suffice to produce significant eddy compensation. As it stands, we think that the respective sentence (now pg.9, ln.4) " ... a more complex 5 definition of the thickness diffusivity (such as in E&G and FMCD) does necessarily amount to an increase of the (parameterized) eddy compensation relative to the original pragmatic choice ..." is correct (i.e. this is what we see in our admittedly coarse-resolution model configurations).

**3) Fig 7c shows different rates of decline in oceanic carbon uptake in the four different experiments performed. I think the linear slopes over years 20-70 should be calculated and compared. This will produce some change between the E&G (blue) slope and the CON and FMCD slopes that is about 20% as large as the slope change in the IRON (green) slope. Is a 20% change "rather robust" as described on pg 9 l 19? It is also unfair to the IRON simulation to say it has the wrong sign of air-sea carbon fluxes (pg 9 l 27), because if the experiment were extended another 10 years, then the sign of the IRON curve in Fig 7c would almost certainly be negative. A better comparison would be the linear slope values. Should spatial maps of the oceanic carbon uptake changes be shown?**

**-A:** Trends are now calculated (pg. 10, Tab. 2) and discussed on more quantitative grounds (pg. 9, ln. 25).

As concerns the presentation of spatial maps of the oceanic carbon uptake - we decided not to add them to the manuscript. Note however that they are now accessible to the public because we included them in our response to the reviewer (http://editor.copernicus.org/index.php/bg-2016-460-AC1.pdf?_mdl=msover_md&_jrl=11&_lcm=oc108lcm109w&_acm=get_comm_file&_ms=55564&c=118347&salt=16528897282134400271).

**4) Pg 11, l 1. A caveat of the present results is that the horizontal resolution of the ocean model is very coarse at 3 deg. Most climate models use a resolution of 1 deg or finer. At NCAR, we now rarely use our 3 deg ocean model because it just doesn't have enough resolution to represent several aspects of the ocean circulation, including the Southern Ocean. I would like to see a comparison like this using 1 deg resolution ocean models to see whether the present conclusions hold, because comparisons with 0.1 deg ocean models with biogeochemistry are still a few years away.**

**-A:** We agree that there is a caveat and we voice the respective information in the revised version of the manuscript more prominently (pg. 11, ln. 20).

As concerns the comparison with higher-resolution models: we are currently working on a 0.1 deg configuration with full biogeochemistry: c.f. http://89.27.255.63/?page_id=90 and https://www.youtube.com/channel/UCnuABRT7qWGgM6bvMzLpr6A and we hope that we can present the respective comparison soon in an additional publication.

**5) Figs 8-10. I would prefer to see observations and then the model minus observations differences, especially in the SSTs in Fig 8.**

**-A:** We show now model - minus observation in Fig. 8 (SST, pg. 23). As for the biogeochemical species (Fig. 9-10) we decided to stick to the original presentation for no other reason than that we feel that this is the more "wide-spread way" in the biogeochemical modeling community. We agree, however, that there are good reasons to go beyond "wide-spread ways"!

**Pg 12, l 2. I disagree. Figs 1, 3 and 5 clearly show that the FMCD choice has a better spatial representation of eddy kinetic energy compared to observations. It also shows a much stronger eddy compensation, which is more in line with eddy-resolving model results. I think it looks a much better choice than E&G or a constant: it really is about time to go beyond using a constant GM coefficient in global climate models.**

**-A:** O.K. We pushed too far in the appendix. In the revised version of the manuscript we will remove the sentence " ... This, in its turn, suggests that the simulated sensitivities of any of our configurations towards changes in the Southern Ocean, are equally likely" (c.f. pg.12, ln. 20).

**1) Pg 1, l 21. The changes in the Southern Hemisphere atmosphere have been driven by changes in the ozone hole as well as by greenhouse gases: Polvani et al (2011), J. Climate, 24, 795.**

**-A:** We added the respective information (and citation) on pg. 1, ln. 21.

**2) Pg 2, l 7. There is also recent evidence that the Southern Ocean carbon sink has been "reinvigorated": Landschutzer et al (2015), Science, 349, 1221.**

**-A:** We added the respective information (and citation) on pg. 2, ln. 9.

**3) Pg 5, l 10-12. There aren't observations of the Southern Ocean MOC, and Bryan et al (2014) should also be referenced here.**

**-A:** We add the respective information (and citation) on pg. 5, ln. 14.

**4) Pg 5, l 28. Coriolis.**

**-A:** O.K.

**5) Pg 7, l 2. Rationale.**

**-A:** O.K.

**6) Pg 8, l 26. Respective.**

**-A:** O.K.

**7) Pg 10, l 8. Reference Swart et al (2014), Biogeosciences, 11, 6107.**

**-A:** We added this reference to the revised version on pg. 10, ln. 21.
* * *
* * *
**Point-by-point - corrections triggered by Review #2; RC2:**

**Pg. 2, Lines 5-7: I would cite more recent studies here and include recent analyses of observations. Up to the mid 2000s there is evidence from models (e.g., Le Quéré et al., 2007; Lovenduski et al., 2007) and observations (please cite Landschützer et al., 2015) that Southern Ocean carbon uptake may have slowed relative to the expected increase due to the increase in atmospheric CO2. More recent observational studies (please cite Landschützer et al., 2015; Munro et al., 2015; and Xue et al., 2015) suggest that the sink may have strengthened over the last decade.**

**-A:** We added the respective information/references on pg. 2, ln.9.

**Pg. 2, Line 8: I would say something more general like the "the link between variability in surface winds and Southern Ocean carbon uptake remains inconclusive"**

**-A:** Changed on pg. 2, ln. 10.

**Pg. 2, Lines 16-22: I would also mention current observational/model studies that have examined carbon uptake associated with mesoscale eddies within the Southern Ocean (please cite Song et al., 2016). This paper includes an analysis of the Drake Passage Timeseries which represents the densest dataset of pCO2 observations within the ACC. Observations are compared to results from a high-resolution (approximately 0.1 degree) simulation of the Southern Ocean region surrounding the Drake Passage. Both observations and model output indicate how a shifting balance of physical and biogeochemical processes drive air-sea carbon flux during different seasons and gives important context to the complexity of the topic presented here.**

**-A:** Song et al., 2016 is cited now on pg. 11, ln. 25.

**Figures: Fig. 7 is the most important in the paper particularly Fig. 7c. I think it would be helpful to include a Table summarizing these results with the linear rate of decrease in C uptake with uncertainty over the 50 years of increased winds. Alternatively, you could present the difference in C uptake with uncertainty between the last twenty years of spin-up and the last five or ten years of increase winds (i.e., years 46-50 or 41-50).**

**-A:** We added a table on pg. 10 and discuss it on pg. 9, ln. 25.

**Figs. 8-11 might be more appropriate in a supplemental information section if allowed so that the reader focuses on the figures most important to the overall story.**

**-A:** We want to keep the information in the appendix.

**TECHNICAL COMMENTS:**
**Pg. 2, Line 2 ...**

**-A:** We applied all your corrections in the revised version of the manuscript. Thank you for combing through so thoroughly.